# Improvement of Statistical Performance of Ordinal Multiscale Entropy Techniques Using Refined Composite Downsampling Permutation Entropy

**DOI:** 10.3390/e23010030

**Published:** 2020-12-28

**Authors:** Antonio Dávalos, Meryem Jabloun, Philippe Ravier, Olivier Buttelli

**Affiliations:** Laboratoire Pluridisciplinaire de Recherche en Ingénierie des Systèmes, Mécanique, Énergétique (PRISME), University of Orléans, 45100 Orléans, France; meryem.jabloun@univ-orleans.fr (M.J.); philippe.ravier@univ-orleans.fr (P.R.); olivier.buttelli@univ-orleans.fr (O.B.)

**Keywords:** multiscale permutation entropy, downsampling, bearing faults

## Abstract

Multiscale Permutation Entropy (MPE) analysis is a powerful ordinal tool in the measurement of information content of time series. MPE refinements, such as Composite MPE (cMPE) and Refined Composite MPE (rcMPE), greatly increase the precision of the entropy estimation by modifying the original method. Nonetheless, these techniques have only been proposed as algorithms, and are yet to be described from the theoretical perspective. Therefore, the purpose of this article is two-fold. First, we develop the statistical theory behind cMPE and rcMPE. Second, we propose an alternative method, Refined Composite Downsampling Permutation Entropy (rcDPE) to further increase the entropy estimation’s precision. Although cMPE and rcMPE outperform MPE when applied on uncorrelated noise, the results are higher than our predictions due to inherent redundancies found in the composite algorithms. The rcDPE method, on the other hand, not only conforms to our theoretical predictions, but also greatly improves over the other methods, showing the smallest bias and variance. By using MPE, rcMPE and rcDPE to classify faults in bearing vibration signals, rcDPE outperforms the multiscaling methods, enhancing the difference between faulty and non-faulty signals, provided we apply a proper anti-aliasing low-pass filter at each time scale.

## 1. Introduction

Information entropy, as described by the seminal paper by Shannon [1], provides a measure of information content. A high entropy level implies a high degree of “unpredictability” in a system. Conversely, a system whose each subsequent state can be easily predicted from the previous ones is said to have low entropy. Shannon defined this measure in the context of communications, by defining and measuring entropy in a digital binary array. Nonetheless, any time series can be analyzed using this type of technique, as long as the “states” of the signal are properly defined in advance.

In particular, ordinal entropy methods have been extensively used for engineering [2], financial [3], and biomedical [4] applications. Ordinal methods, like Permutation Entropy (PE) [5] have the advantage of noise-robustness and invariance to non-linear monotonous transformations. Furthermore, the implementation of the multiscale approach [6] leads to the Multiscale Permutation Entropy (MPE) [7] technique. This allows us to search for information contained in long trends within a signal, which can be lost if we analyze its raw data points directly. PE and MPE relies on the pattern counts inside a signal, which itself is just a sample series from a wider phenomenon. Therefore, the entropy content measured is not a deterministic measure, but an estimator.

Several refinements to MPE have been proposed [8,9], providing a better precision over the original MPE formulation. These approaches rely on the composite multiscale techniques by Wu [10], which is able to extract more information from a signal than classical multiscaling.

In our previous work, we provided a theoretical expression of the MPE bias [11] and variance [12]. Furthermore, we proved the MPE statistic is approximately efficient, approaching the Cramér-Rao lower bound. Nonetheless, composite ordinal multiscale techniques have not been properly addressed in the literature from the statistical point of view.

Therefore, the aim of this article is twofold. First, we will develop the statistical properties of MPE refinements, Composite MPE (cMPE) [8], and Refined Composite MPE (rcMPE) [13], in order to properly characterize their performance from the theoretical perspective. Second, and most importantly, we propose an alternative to the composite multiscaling process, based on downsampling, to further refine the precision of these ordinal permutation entropy methods. We will provide the theoretical statistical characterization of this proposal, as well as examples of its application in real-life scenarios

In Section 2, we will lay out the necessary background theory, including the formal definitions of PE, MPE, and their refinements. In Section 3, we will develop the proper mathematical expressions for the expected value, bias and variance of the refined MPE methods. Furthermore, we will propose the alternative Composite Downsampling, as well as their corresponding statistical characterization. Lastly, in Section 4 we will test the performance of each of the methods discussed using uncorrelated noise signals, as well as real signals from bearing fault recordings [14].

## 2. Theoretical Framework

This section will cover the necessary background for the PE methods. We will present the original formulation of PE in [5], as the MPE proposal by [7]. Subsequently, we will provide a brief summary of the MPE statistical moments, which we originally developed in [11,12].

### 2.1. Permutation Entropy

For a signal $x={\left[x_{1},x_{2},\ldots ,x_{N}\right]}^{T}$ with *N* data samples, we define ordinal patterns of embedded dimension *d* as any possible ordinal permutation between *d* points of the signal. For example, for d=2, only two possible ordinal patterns exist: $x_{t}<x_{t+1}$ and $x_{t}>x_{t+1}$; if dimension d=3, we could obtain pattern any of the six possible ordinal permutations of $x_{t}$, $x_{t+1}$, and $x_{t+2}$ (for example, $x_{t}<x_{t+1}<x_{t+2}$). In general, for embedded dimension *d*, there are d! possible patterns.

For the aforementioned signal, we will assume no particular structure or statistical properties and establish that said signal must be uniformly sampled as the only restriction. Since this assumption implies that no further information is known, we can only estimate the probability of each pattern by measuring the pattern counts inside the signal; in other words, we measure the cardinality of each pattern i=1,⋯,d! [5],
(1)$$y_{i}=\#\left\{n|\;n<N-\left(d-1\right)\tau ,\;\left(x_{n},x_{n+\tau }\cdots ,x_{n+(d-1)\tau }\right)\;\mathrm{has}\;\mathrm{pattern}\;i\right\},$$
where $y_{i}$ is the number of patterns of type *i* in the signal x, and $\tau \in N^{+}$ is a downsampling parameter, as defined by [15]. Some examples of possible ordinal patterns are shown in Figure 1. We can use $y_{i}$ to build an estimate of the pattern probability ${\hat{p}}_{i}$:(2)$${\hat{p}}_{i}=\frac{\#\{n|\;n<N-\left(d-1\right)\tau ,\;\left(x_{n},x_{n+\tau }\cdots ,x_{n+(d-1)\tau }\right)\;\mathrm{has}\;\mathrm{pattern}\;i\}}{N-(d-1)\tau },$$
We use the symbol ${\hat{p}}_{i}$ here to denote that this is an estimation of the pattern probability, as opposed to $p_{i}$, which represents the true value.

For a given dimension *d*, the estimate probabilities for all possible patterns *i* form a mass probability distribution function (pmf). Therefore, it is possible to obtain an estimation of the permutation entropy measurement, denoted by $\hat{H}$, using Shannon’s entropy definition [5],
(3)$$\hat{H}=-\sum_{i=1}^{d!}{\hat{p}}_{i}\ln{\hat{p}}_{i}.$$
Here, the value $\hat{H}$ is an estimation, and thus, a statistic. We can also write the normalized version of (Equation 3) as follows,
(4)$$\hat{H}=\frac{\hat{H}}{\ln(d!)}=\frac{-1}{\ln(d!)}\sum_{i=1}^{d!}{\hat{p}}_{i}\ln{\hat{p}}_{i}$$
which guarantees an entropy value between zero and one.

From the algorithmic point of view, PE is easy to implement and fast to compute given a signal x and a dimension *d*. Since the nature of this procedure is ordinal, PE is invariant to nonlinear monotonous transformations [5], which is in turn a desired property when we expect to work with signals containing different amplitudes, noise, or outliers. On the other hand, this robustness works against us if we intent to extract information from the signal’s amplitude.

Another obvious constraint is the length of the signal. For very short signals, the pattern counts in (Equation 2) are not sufficient to provide a precise estimation of the pattern distribution. There are several proposed guidelines for the minimum length required, with the condition N≫d! [5] being the most notable one—this, however, offers no practical guidelines to the proper size of *N*. Other possible length constrain formulations are N≥5d! [16] and N>(d+1)! [17].

This technique, of course, only deals with a single set of data points, τ spaces apart. In order to measure the information content on long-range correlations, it is necessary to introduce the multiscale approach.

### 2.2. Multiscale Permutation Entropy

Using the original signal x, we can construct coarse-grained signals for a fixed time scale *m*. We partition the data points in consecutive, non-overlapping segments of size *m*, computing the average of each segment afterwards and constructing $x^{\left(m\right)}={\left[x_{1}^{\left(m\right)},\cdots ,x_{N/m}^{\left(m\right)}\right]}^{T}$, where each element is,
(5)$$x_{j}^{\left(m\right)}=\frac{1}{m}\sum_{i=m(j-1)+1}^{jm}x_{i},$$
where j∈N and m∈N. The MPE measurement consists on calculating the permutation entropy of $x^{\left(m\right)}$ for different time scales.

As pointed out in [18], this process can be regarded as a two-step procedure: (a) a moving average filter, and (b) a downsampling. The motivation behind the coarse-graining procedure is to capture long-term information, usually lost in the ordinal comparisons between adjacent data points. The assessment of this information can be useful in detecting trends or recurring patterns that are not usually evident in raw data.

However, MPE also has the disadvantage of being sensitive to signal length: as *N* decreases, the estimation of MPE will be less reliable. This effect becomes more pronounced with increasing scales, where the size of the coarse-grained signal decreases by a factor of 1/m. The general condition N/m≫d! must be satisfied.

### 2.3. MPE Statistics

In our previous work [11,12], we developed some of the statistical properties of the MPE, including the expected value and variance. In this section we will briefly summarize these findings.

For mathematical convenience, we will restate the pattern counts (Equation 1) and pattern probability estimations (Equation 2) in vectorial form. For any coarse grained signal $x^{\left(m\right)}$ of length $n_{m}\approx N/m$ at time scale *m* and dimension *d*, we can define the random vector Y of size d! as the vector of pattern counts, and $\hat{p}$ as the vector of pattern probabilities from Equation (Equation 2):(6)$$\begin{array}{ccc} Y & =\left[\begin{array}{c} Y_{1}\\ \vdots \\ Y_{d!} \end{array}\right]=\left[\begin{array}{c} n_{m}p_{1}+\Delta Y_{1}\\ \vdots \\ n_{m}p_{d!}+\Delta Y_{d!} \end{array}\right]=n_{m}p+\Delta Y, & Y∼Mu(n_{m},p) \end{array}$$(7)$$\begin{array}{cc} \hat{p} & =\frac{1}{n_{m}}Y=p+\frac{1}{n_{m}}\Delta Y. \end{array}$$
A realization Y is assumed to follow a multinomial distribution for each of the random variables $Y_{1},\cdots ,Y_{d!}$, will give the cardinality of each pattern count $y_{1},\cdots ,y_{d!}$ defined in (Equation 1). If we further define the vector
(8)$$\begin{array}{c} \hat{l}=\ln\hat{p}, \end{array}$$
then we can rewrite the PE in vectorial form,
(9)$$\hat{H}=H\left(\hat{p}\right)=-\sum_{i=1}^{d!}{\hat{p}}_{i}\ln\left({\hat{p}}_{i}\right)=-\hat{l}{\;}^{T}\hat{p},$$
where *T* is the transpose symbol. In order to obtain the statistical moments of $\hat{H}$, it is convenient to expand this expression into its second order Taylor series approximation [19]:(10)$$H\left(\hat{p}\right)\approx H\left(p\right)-\frac{m}{N}{\left(1+l\right)}^{T}\Delta Y-\frac{1}{2}{\left(\frac{m}{N}\right)}^{2}{\left(p^{\circ -1}\right)}^{T}\Delta Y^{\circ 2}+O\left(\Delta Y^{\circ 3}\right),$$
where ΔY is the random part of the pattern count in (7), and $\Delta Y^{\circ 2}$ is the Hadamard square of ΔY. This expression leads directly to the MPE expected value [11,12],
(11)$$\begin{array}{c} E\left[H\left(\hat{p}\right)\right]\approx H\left(p\right)-\frac{1}{2}\left(d!-1\right)\left(\frac{m}{N}\right) \end{array}$$
and variance [12]
(12)$$\begin{array}{cc} var\left(H(\hat{p})\right) & \;\approx \;\left(\frac{m}{N}\right)l^{\prime T}\Sigma_{p}l\;+\;{\left(\frac{m}{N}\right)}^{2}\left(1^{T}l+d!H\left(p\right)+\frac{1}{2}\left(d!-1\right)\right). \end{array}$$

Here, the term,
(13)$$\begin{array}{cc} \left(\frac{m}{N}\right)l^{T}\Sigma_{p}l & \;=\;\left(\frac{m}{N}\right)\left(\sum_{i=1}^{d!}p_{i}\ln^{2}p_{i}-H^{2}\left(p\right)\right), \end{array}$$
corresponds to the Cramér-Rao lower bound for *H* [12]. This shows that it is still approximately efficient for *H*.

The fact that we have an approximate measure for the bias and variance of *H* allows us to assess the error in our estimation, and properly gauge the precision of our measurements. This is useful when comparing entropy content between two different groups, especially when we have signals of different length.

Although the Cramér-Rao lower bound for the MPE statistic suggests minimum variance, it is still possible to increase the precision of the entropy measurement even further. The composite MPE methods provide a way to increase the number of samples, and thus, reduce the variation of our estimations.

### 2.4. Composite MPE Methods

Composite coarse-graining (CCG) stems from the notion that, for a given time series x and a set time scale *m*, we can build an *m* different coarse signals if we change the starting element for the coarse-graining procedure. Up to this moment, original MPE assumes that the starting point is equal to the first element of the time series ($x_{1}$). At most, we can have an *m* number of difference signals for the same time scale that are similar to one another yet contain slightly different information. We apply the general procedure to build all the possible coarse signals for *m*,
(14)$$x_{k,j}^{\left(m\right)}=\frac{1}{m}\sum_{i=m(j-1)+k}^{jm+(k-1)}x_{i},$$
with k=1,⋯,m for the starting element, which also refers to the $k^{th}$ coarse signal for scale *m*. Applying the procedure in (Equation 14) gives us coarse signals $x_{1}^{\left(m\right)},\cdots ,x_{m}^{\left(m\right)}$ for any given *m*. A schematic example is show in in Figure 2.

This composite approach allows us to access previously unaccounted ordinal patterns in the series, thus increasing their number for the purposes of building the empirical pattern probability distribution. Therefore, by applying the PE method on composite coarse signals, it is possible to partially overcome the length constraints, especially in the context of multiscaling.

In the following subsections, we will outline the most common composite methods: composite MPE (cMPE) [8] and the refined composite MPE (rcMPE) [9].

#### 2.4.1. Composite MPE

Following the composite coarse-graining procedure in Equation (Equation 14) [8] makes it possible to achieve better precision by averaging the MPE result for all the composite signals with the same time scale. Even though this approach was originally named “improved multiscale permutation entropy” [8], we will refer to this procedure as composite MPE (cMPE) due to its shared similarities with composite multiscale entropy—as proposed by Wu et al. [18] using SampEn—in its mathematical approach.

Given the original time series x and embedding dimension *d*, we compute the classical MPE on each of the possible composite coarse signals $x_{1}^{\left(m\right)},\cdots ,x_{m}^{\left(m\right)}$ for each time scale *m* to obtain the cMPE. The cMPE value is the average of each of the resulting MPE measurements of all *m* coarse signals:(15)$$H_{cM}\left({\hat{p}}^{\left(m\right)}\right)=\frac{1}{m}\sum_{k=1}^{m}H_{k}\left({\hat{p}}^{\left(m\right)}\right),$$
where $H_{k}\left({\hat{p}}^{\left(m\right)}\right)$ are the k=1,⋯,m possible MPE values. The approach of this method relies on reducing the variance by taking the average of multiple MPE measurements, each calculated on a coarse composite signal.

The authors applied the cMPE technique both in synthetic signals and in EEG recordings, obtaining improved results in terms of stability (variance), which improves the use of ordinal entropy technique for the purposes of diagnosis and biomedical signal classification.

#### 2.4.2. Refined Composite MPE

Originally proposed by Humeau-Heutier et al. [9], rcMPE approaches composite coarse signals through a different mechanism. Instead of using the average MPE, this method counts all the ordinal patterns contained in composite coarse signals for a given scale *m*. This results in a single pattern probability estimation (Equation 16), which is used thereafter to obtain the entropy measurement. Computing rcMPE requires us to first take the average estimation for each pattern probability,
(16)$$\begin{array}{cc} {\hat{\bar{p}}}^{\left(m\right)}\; & =\left[\begin{array}{c} {\hat{\bar{p}}}_{1}^{\left(m\right)}\\ {\hat{\bar{p}}}_{2}^{\left(m\right)}\\ \vdots \\ {\hat{\bar{p}}}_{d!}^{\left(m\right)} \end{array}\right]=\left[\begin{array}{c} \frac{1}{m}\sum_{k=1}^{m}{\hat{p}}_{k,1}^{\left(m\right)}\\ \frac{1}{m}\sum_{k=1}^{m}{\hat{p}}_{k,2}^{\left(m\right)}\\ \vdots \\ \frac{1}{m}\sum_{k=1}^{m}{\hat{p}}_{k,d!}^{\left(m\right)} \end{array}\right], \end{array}$$
where the pattern probability estimator ${\hat{p}}_{k,i}^{\left(m\right)}$ is obtained using Equation (Equation 2) for each composite coarse signal *k*, from Equation (Equation 14). At this point, it is enough to make a single MPE computation over this pattern probability,
(17)$$H_{rcM}\left({\hat{\bar{p}}}^{\left(m\right)}\right)=-\sum_{i=1}^{d!}{\hat{\bar{p}}}_{i}^{\left(m\right)}\ln{\hat{\bar{p}}}_{i}^{\left(m\right)}=-\hat{\bar{l}}{\;}^{T}\hat{\bar{p}},$$
following the same procedure as the original MPE definition (Equation 4) and (Equation 8), using ${\hat{\bar{p}}}^{\left(m\right)}$ instead of ${\hat{p}}^{\left(m\right)}$.

When applied to laser Doppler flowmetry and bearing fault signals [9], rcMPE shows not only a reduced variance compared to the MPE and cMPE algorithms, but it is also less affected by the length constrains explained in Section 2.2. In Section 3.1, we will expand the analysis of the MPE statistical properties—which we previously did in [12]—to the cMPE and rcMPE methods, in order to understand the source of this improved performance.

## 3. Methods

In this section, we will explicitly develop the statistical moments of the cMPE and rcMPE, as well as discussing some of the implications of these results.

### 3.1. cMPE and rcMPE Statistics

In this section we will expand the results from [11] and variance [12] in order to properly describe these statistics using the CCG approach. Here, we derive the theoretical statistic development which supports the increased precision of the cMPE and rcMPE found by [9].

#### 3.1.1. cMPE Moments

In the case of cMPE statistics, we can obtain the expected value of $\hat{H}$ using Equation (Equation 15),
(18)$$E\left[H_{cM}\left({\hat{p}}^{\left(m\right)}\right)\right]=\frac{1}{m}\sum_{k=1}^{m}E\left[H_{k}\left({\hat{p}}^{\left(m\right)}\right)\right]\approx H\left(p^{\left(m\right)}\right)-\frac{1}{2}\left(d!-1\right)\left(\frac{m}{N}\right).$$
This mean value is exactly the same as the classical MPE mean (Equation 11), and therefore, we should expect the same results in average. In the case of the variance, if we suppose all ${\hat{H}}_{k}\left({\hat{p}}^{\left(m\right)}\right)$ are independent for k=1,⋯,m, the general expression of the cMPE variance can be written as,
(19)$$\begin{array}{cc} var\left(H_{cM}\left({\hat{p}}^{\left(m\right)}\right)\right) & =\frac{1}{m^{2}}var\left(\sum_{k=1}^{m}H_{k}\left({\hat{p}}^{\left(m\right)}\right)\right)\\ \; & =\frac{1}{m^{2}}\sum_{k=1}^{m}var\left(H_{k}\left({\hat{p}}^{\left(m\right)}\right)\right)+\frac{1}{m^{2}}\overset{m}{\underset{k_{1}\neq k_{2}}{\sum \sum }}cov\left(H_{k_{1}}\left({\hat{p}}^{\left(m\right)}\right),H_{k_{2}}\left({\hat{p}}^{\left(m\right)}\right)\right)\\ \; & =\frac{1}{m}var\left(H_{k}\left({\hat{p}}^{\left(m\right)}\right)\right)+\frac{1}{m^{2}}\overset{m}{\underset{k_{1}\neq k_{2}}{\sum \sum }}cov\left(H_{k_{1}}\left({\hat{p}}^{\left(m\right)}\right),H_{k_{2}}\left({\hat{p}}^{\left(m\right)}\right)\right),\;\forall k\\ \; & \approx \left(\frac{1}{N}\right){l^{\left(m\right)}}^{T}\Sigma_{p}^{\left(m\right)}l^{\left(m\right)}\;+\;\left(\frac{1}{N}\right)\left(\frac{m}{N}\right)\left(1^{T}l^{\left(m\right)}+d!H\left(p^{\left(m\right)}\right)+\frac{1}{2}\left(d!-1\right)\right), \end{array}$$
where $k_{1}=1,\cdots ,m$ and $k_{2}=1,\cdots ,m$. Note that the first term is equal to the CRLB from Equation (Equation 13).The measure of equality should be reached when the $H_{k}\left({\hat{p}}^{\left(m\right)}\right)$ are not correlated.

When comparing MPE and cMPE, it should not come as a surprise that the expected value (Equation 18) does not change. Nonetheless, the variance in Equation (Equation 19) is indeed reduced by a factor of 1/m from Equation (Equation 12). This has a visible effect on the polynomial approximation (Equation 12), reducing the degree by one. Now the first element is constant with respect to *m*, and the second term is linear. Equation (Equation 19) provides a benchmark for the minimum variance the cMPE can obtain in the presence of uncorrelated coarse signals.

#### 3.1.2. rcMPE Moments

For the rcMPE statistics, the explicit representation of the first two moments $H_{rcM}\left({\hat{p}}^{\left(m\right)}\right)$ require further explanation. First, we modify the original multinomial pattern count expression from Equations (Equation 6) and (7), such that we can describe each one of the composite coarse-grained signals for scale *m*,
$$\begin{array}{ccc} Y_{k} & =\left[\begin{array}{c} Y_{k,1}\\ Y_{k,2}\\ \vdots \\ Y_{k,d!} \end{array}\right]=\left[\begin{array}{c} n_{m}p_{1}+\Delta Y_{k,1}\\ n_{m}p_{2}+\Delta Y_{k,2}\\ \vdots \\ n_{m}p_{d!}+\Delta Y_{k,d!} \end{array}\right]=n_{m}p+\Delta Y_{k}, & ∼Mu(n_{m},p) \end{array}$$
(20)$${\hat{p}}_{k}=\frac{1}{n_{m}}Y_{k}=p+\frac{1}{n_{m}}\Delta Y_{k}.$$
The random variable $Y_{k}$ represents the counts for each possible ordinal pattern and $n_{m}\approx N/m$. We then obtain the average of all the pattern counts along all the composite signals:(21)$$\begin{array}{cc} \bar{Y} & =\frac{1}{m}\sum_{k=1}^{m}Y_{k}=\frac{1}{m}\sum_{k=1}^{m}\frac{N}{m}p+\frac{1}{m}\sum_{k=1}^{m}\Delta Y_{k}\\ \bar{Y} & =\frac{N}{m}p+\frac{1}{m}\sum_{k=1}^{m}\Delta Y_{k} \end{array}$$
If we rearrange Equation (Equation 21) we can obtain an expression for the expected value of p,
(22)$$\begin{array}{cc} \hat{\bar{p}} & =p+\frac{1}{N}\sum_{k=1}^{m}\Delta Y_{k}. \end{array}$$

Rewriting the refined composite technique in such a way is revealing: Equation (Equation 22) is identical to the probability estimation in (Equation 16) and this formulation shows explicitly that the estimation is now independent of the time scale value in *m*. However, we cannot guarantee that $\bar{Y}$ has a multinomial distribution, since all the $\Delta Y_{k}$ in (Equation 21) are not necessarily uncorrelated.

If we use (Equation 22) in the classical MPE Taylor series approximation (Equation 10), we obtain
(23)$$H_{rcM}\left({\hat{\bar{p}}}^{\left(m\right)}\right)\approx H_{rcM}\left(p\right)-\frac{1}{N}{\left(1+l\right)}^{T}\sum_{k=1}^{m}\Delta Y_{k}-\frac{1}{2}{\left(\frac{1}{N}\right)}^{2}{\left({\left(p^{\left(m\right)}\right)}^{\circ -1}\right)}^{T}\sum_{k=1}^{m}\Delta Y_{k}^{\circ 2},$$
where l=ln(p). Obtaining the expected value of (Equation 23) is enough to also obtain the mean rcMPE. We can clearly write the expected value as,
(24)$$E\left[H_{rcM}\left({\hat{\bar{p}}}^{\left(m\right)}\right)\right]\approx H\left(p^{\left(m\right)}\right)-\frac{1}{2}\left(d!-1\right)\left(\frac{1}{N}\right).$$

The variance has some additional complication. Since $\bar{Y}$ is not strictly multinomial, the moments of $\sum_{k=1}^{m}\Delta Y_{k}$ do not follow the corresponding moments. If we take the naïve approach of assuming all $\Delta Y_{k}$ are mutually independent, we can write the rcMPE variance as,
(25)$$\begin{array}{c} var\left(H_{rcM}\left({\hat{\bar{p}}}^{\left(m\right)}\right)\right)\approx \left(\frac{1}{N}\right){l^{\left(m\right)}}^{T}\Sigma_{p}^{\left(m\right)}l^{\left(m\right)}\;+\;{\left(\frac{1}{N}\right)}^{2}\left(1^{T}l^{\left(m\right)}+d!H\left(p^{\left(m\right)}\right)+\frac{1}{2}\left(d!-1\right)\right). \end{array}$$

As we can observe from (Equation 24) and (Equation 25), we have eliminated the *m* dependence, so we now have a constant value of the mean, bias, and variance, which relies solely on the signal’s original length and the embedded dimension. Thanks to this refinement, it is possible for us now to explore higher *m* values without worrying about loss of precision due to signal length reduction. Nonetheless, we still have to address the deviation from the variance in Equation (Equation 25), where we have no guarantee of independent pattern counts across all composite signals.

#### 3.1.3. Composite MPE Statistics—Discussion

At this point, we must remark one of the main shortcomings in this approach that is not mentioned in the existing literature. If we compare the same elements from different coarse signals at a given *m* and revisit the definition in (Equation 14), we observe that the elements between signals share information and that segments from different coarse signals overlap. For example, a closer look at Figure 2 reveals that the first elements of the coarse signals $x_{1}^{\left(m\right)}$ and $x_{2}^{\left(m\right)}$ are
$$\begin{array}{cc} x_{k=1,j=1}^{\left(m=3\right)} & =\frac{1}{m}\left(x_{1}+x_{2}+x_{3}\right)\\ x_{k=2,j=1}^{\left(m=3\right)} & =\frac{1}{m}\left(x_{2}+x_{3}+x_{4}\right). \end{array}$$

Since elements $x_{2}$ and $x_{3}$ appear in both signals, all the information that we could measure from coarse signals $x_{1}^{\left(m\right)}$ and $x_{2}^{\left(m\right)}$ will have some level of redundancy. This becomes more evident as we increase the value of *m*, and consequently, the number of shared elements increases.

This redundancy is bound to create cross-correlation between coarse signals, even if the original signal is uncorrelated. This effect will influence the overall MPE estimators that rely on composite signals, possibly resulting in an increased variance due to this redundant information. Here, we refer to this effect as an artifact cross-correlation: the presence of correlation between coarse signals originating from shared elements and not from the inherent dynamics of the original signal.

The existence of artifact cross-correlation implies that our measurement of variance in (Equation 19) and (Equation 25) will not be satisfied. For these expressions to be true, we must guarantee that no artifact cross-correlations appear between the composite coarse signals as a consequence of the CCG procedure.

### 3.2. Composite Downsampling

Instead of fully characterizing this artifact cross-correlation effect, a complex and time-consuming mathematical endeavor, we will present an alternative that is exempt of this redundancy from the beginning: composite downsampling.

Downsampling in the context of PE is not a new concept, as it is shown in Equation (Equation 2) by including the downsampling parameter τ [15]. Moreover, a form of composite downsampling has already been proposed in [20] in the context of the measurement of information content in chaotic signals. This approach has been replaced by the cMPE approach proposed in [8], since the application of a downsampling instead of a moving average filter incurs in the risk of aliasing. Nonetheless, refs [8,20] do not discuss the artifact cross-correlation effect, nor the statistical properties of these approaches. Thus, revisiting these concepts from this perspective is worth the effort.

From the cardinal point of view, the coarse-graining procedure represents a better smoothing filter than a simple downsampling procedure, since the former incorporates more signal information and the latter implies a loss of resolution as a trade-off. Nonetheless, both procedures behave similarly for the purpose of ordinal patterns.

For this purpose, the composite downsampled signals [20], are defined as follows for k=1,⋯,τ:(26)$$x_{k,j}^{\left(\tau \right)}=x_{k+\tau (j-1)}.$$

Changing the starting element *k* allows us to obtain a τ number of downsampled signals from the original signal x. This implies no information loss, since all the elements in x are still present in the composite signals $x_{k}^{\left(\tau \right)}$ (see Figure 3). Additionally, since we can also appreciate that the resulting signals have no elements in common, we know that the artifact cross-relation effect will not be present. This is justified if we regard the process (Equation 26) as a systematic sampling, where each downsampled signal is a sample of the “population” signal x, with the constraint that no sample groups share mutual elements.

For a signal x of length *N*, the MPE estimator is expected to have the moments found in Equations (Equation 11) and (Equation 12). Both the expected value and the variance depend on time scale *m*, both implicitly (by means of ${\hat{p}}^{\left(m\right)}$) and explicitly. It is also worth mentioning that these moments are heavily dependent on the signal length *N* and the embedding dimension *d*.

Since the downsampling procedure also reduces the signal length in a similar fashion, it stands to reason that applying a classical downsampling procedure with any given τ value will present the moments as follows:(27)$$\begin{array}{cc} E[H\left({\hat{p}}^{\left(\tau \right)}\right)] & \approx H\left(p^{\left(\tau \right)}\right)-\frac{1}{2}\left(d!-1\right)\left(\frac{\tau }{N}\right) \end{array}$$(28)$$\begin{array}{cc} var\left(H({\hat{p}}^{\left(\tau \right)})\right) & \;\approx \;\left(\frac{\tau }{N}\right)l^{\prime (\tau )}\Sigma_{p}^{\left(\tau \right)}l^{\left(\tau \right)}\;+\;{\left(\frac{\tau }{N}\right)}^{2}\left(1^{\prime }l^{\left(\tau \right)}+d!H\left(p^{\left(\tau \right)}\right)+\frac{1}{2}\left(d!-1\right)\right). \end{array}$$

Given that the only explicit change is switching from τ instead of *m*, we can capitalize on the MPE theory and apply it to the downsampling case.

Certainly, we cannot expect to obtain the same pattern probabilities from these two procedures due to the fact that, in general, $p^{\left(\tau \right)}\neq p^{\left(m\right)}$. Still, we expect to get the exact same pattern distribution (i.e., uniform) for both procedures in some specific circumstances, such as a signal consisting of uncorrelated noise.

In the following subsections we will develop the theoretical framework for two new entropy measurements: composite downsampling permutation entropy (cDPE) and refined composite downsampling permutation entropy (rcDPE).

#### 3.2.1. Composite Downsampling PE

Similarly to the case of cMPE, utilizing the composite downsampling procedure (Equation 26) allows us to define cDPE as:(29)$$H_{cD}\left({\hat{p}}^{\left(\tau \right)}\right)=\frac{1}{\tau }\sum_{k=1}^{\tau }H_{k}\left({\hat{p}}^{\left(\tau \right)}\right).$$

This definition is identical to the one proposed in [20], named Modified MPE (MMPE). It is by means of the downsampling procedure (Equation 26) that we can deduce that all $H_{k}\left({\hat{p}}^{\left(\tau \right)}\right)$ are independent for k=1,⋯,τ. Therefore, we find the traditional moments for the mean of $\hat{H}$, namely
(30)$$E\left[H_{cD}\left({\hat{p}}^{\left(\tau \right)}\right)\right]=\frac{1}{\tau }\sum_{k=1}^{\tau }E\left[H_{k}\left({\hat{p}}^{\left(\tau \right)}\right)\right]\approx H\left(p^{\left(\tau \right)}\right)-\frac{1}{2}\left(d!-1\right)\left(\frac{\tau }{N}\right),$$
and
(31)$$\begin{array}{cc} var\left(H_{cD}\left({\hat{p}}^{\left(\tau \right)}\right)\right) & =\frac{1}{\tau^{2}}var\left(\sum_{k=1}^{\tau }H_{k}\left({\hat{p}}^{\left(\tau \right)}\right)\right)\\ \; & =\frac{1}{\tau^{2}}\sum_{k=1}^{\tau }var\left(H_{k}\left({\hat{p}}^{\left(\tau \right)}\right)\right)\\ \; & =\frac{1}{\tau }var\left(H_{k}\left({\hat{p}}^{\left(\tau \right)}\right)\right),\;\forall k\\ \; & \approx \left(\frac{1}{N}\right){l^{\left(\tau \right)}}^{T}\Sigma_{p}^{\left(\tau \right)}l^{\left(\tau \right)}\;+\;\left(\frac{1}{N}\right)\left(\frac{\tau }{N}\right)\left(1^{T}l^{\left(\tau \right)}+d!H\left(p^{\left(\tau \right)}\right)+\frac{1}{2}\left(d!-1\right)\right). \end{array}$$

Once again, the cDPE expected value (Equation 30) does not change with respect to classical MPE, and we still have a downward bias whose only dependencies are the embedding dimension *d*, signal length *N*, and the downsampling parameter τ. Conversely, the cDPE variance (Equation 31) is reduced by a factor of 1/τ in this case, and we expect it to be close to (Equation 31) due to the lack of an artifact cross-correlation by virtue of the definition shown in (Equation 26). This implies an improvement over the cMPE variance (Equation 19), where the cross-correlations will invariably reduce the estimator’s precision.

Although the definition in (Equation 29) is not new [20], we present here the mathematical expressions for the mean and variance for the first time. As previously discussed in [8], this particular procedure has the shortcoming of being prone to aliasing due to the downsampling procedure been applied without any previous filtering. Nonetheless, the particular advantage of not having redundant information has not been previously addressed in the literature.

#### 3.2.2. Refined Composite Downsampling PE

In this section, here we propose the rcDPE algorithm as a new alternative to both the MPE methods, and the cDPE/MMPE algorithm. Here we combine the composite downsampling approach with the refined algorithm proposed by [9]. By following the same reasoning, rcDPE takes the average pattern probability distribution from all the downsampled signals for the parameter τ.

As with (Equation 16), we proceed to define the probability vector,
(32)$$\begin{array}{cc} {\hat{\bar{p}}}^{\left(\tau \right)}\; & =\left[\begin{array}{c} {\hat{\bar{p}}}_{1}^{\left(\tau \right)}\\ {\hat{\bar{p}}}_{2}^{\left(\tau \right)}\\ \vdots \\ {\hat{\bar{p}}}_{d!}^{\left(\tau \right)} \end{array}\right]=\left[\begin{array}{c} \frac{1}{\tau }\sum_{k=1}^{\tau }{\hat{p}}_{k,1}^{\left(\tau \right)}\\ \frac{1}{\tau }\sum_{k=1}^{\tau }{\hat{p}}_{k,2}^{\left(\tau \right)}\\ \vdots \\ \frac{1}{\tau }\sum_{k=1}^{\tau }{\hat{p}}_{k,d!}^{\left(\tau \right)} \end{array}\right]. \end{array}$$
where the pattern probability estimator ${\hat{p}}_{k,i}^{\left(\tau \right)}$ is obtained using Equation (Equation 2) for each composite downsampled signal *k*. At this point, it suffices to apply a single PE computation to produce this pattern probability.

For the composite downsampling procedure, we present rcDPE by using the exact same approach as before,
(33)$$H_{rcD}\left({\hat{\bar{p}}}^{\left(\tau \right)}\right)=-\sum_{i=1}^{d!}{\hat{\bar{p}}}_{i}^{\left(\tau \right)}\ln{\hat{\bar{p}}}_{i}^{\left(\tau \right)},$$
and following the same procedure as the original MPE definition, using ${\hat{\bar{p}}}^{\left(\tau \right)}$ instead of ${\hat{\bar{p}}}^{\left(m\right)}$.

Once again we can enunciate the explicit moments of rcDPE, for the downsampled signals display no artifact cross-correlation. By following the procedure outlined for rcMPE in Section 3.1, we obtain the rcDPE expected value,
(34)$$E\left[H_{rcD}\left({\hat{\bar{p}}}^{\left(\tau \right)}\right)\right]\approx H\left(p^{\left(\tau \right)}\right)-\frac{1}{2}\left(d!-1\right)\left(\frac{1}{N}\right),$$
and the rcDPE variance,
(35)$$\begin{array}{c} var\left(H_{rcD}\left({\hat{\bar{p}}}^{\left(\tau \right)}\right)\right)\approx \left(\frac{1}{N}\right){l^{\left(\tau \right)}}^{T}\Sigma_{p}^{\left(\tau \right)}l^{\left(\tau \right)}\;+\;{\left(\frac{1}{N}\right)}^{2}\left(1^{T}l^{\left(\tau \right)}+d!H\left(p^{\left(\tau \right)}\right)+\frac{1}{2}\left(d!-1\right)\right). \end{array}$$

In contrast to rcMPE, we do not expect in this case for the expected value (Equation 34) and its variance (Equation 35) to raise above the aforementioned mathematical expressions, since the artifact cross-correlation is not present. We still have the advantage of taking the explicit τ parameter dependence out of the equation, which heavily suggests a stable behavior across $\tau \in N^{+}$.

## 4. Results and Discussion

Here, we will proceed to test the statistical theory we developed in the previous section, in order to assess the performance of the aforementioned entropy methods. In order to do this, we will perform the MPE, cMPE, rcMPE, cDPE and the proposed rcDPE techniques over test signals. We will first analyze uncorrelated white noise, in order to characterize the artifact cross-correlation effect on each technique. We will then proceed to apply these entropy methods to real signals (bearing fault recordings [14]), in order to compare their performance for classification purposes. We will also take a close look at the effect on the frequency content on each method.

### 4.1. Uncorrelated Signals

In this section, we will apply all the entropy techniques discussed on simulated white Gaussian noise, with 500 signals of length N=1000 for d=3. Each signal was set to have an expected value of μ=0 and variance $\sigma^{2}=1$, although this works for any chosen variance, due to the PE invariance to amplitude, discussed in Section 2.1. We will perform the analysis for the first 20 time scales. We will set m=τ in order to compare the coarse-graining and the downsampling entropy techniques.

We can see the mean PE results in Figure 4a for all methods, and Figure 4b for the rcMPE and rcDPE methods only. As we expected from Equations (Equation 11), (Equation 18) and (Equation 30), the MPE, cMPE and cDPE all present the same downward trend with increasing scale. This, however, do not completely follow the predicted values (shown by the black dotted line). One possible explanation comes from the fact that the pattern counts are not strictly independent between each other [21]. Nonetheless, this still shows the original MPE, as well as the composite methods, having a time scale dependency, even for uncorrelated white noise, where no such trend is expected. As it is clearly seen in Figure 4b, even the rcMPE presents a reduced downward drift, not accounted by Equation (Equation 24). This effect is due to the artifact cross-correlation between coarse-grained segments. We support this claim by observing the behavior of rcDPE, which is the only method whose entropy value is completely independent of time scale, as shown in Equation (Equation 34). Since (Equation 24) and (Equation 34) are functionally the same equation, the only difference comes from the artifact cross-correlation effect. The non-independent patterns consideration [21], on the other hand, can explain the difference between the measured rcDPE and the theoretical horizontal PE line shown in Figure 4b. Finally, Figure 4c shows the PE bias for each method.

Figure 4d–f show the variance obtained from the PE methods over simulated uncorrelated noise signals. As we can see from Figure 4d, the original MPE presents the highest variance from all the methods, which confirms the need for composite and refined composite MPE instead of the classical algorithm. Moreover, the MPE variance increases quadratically (Equation 12), as expected from our previous work [12] for uncorrelated signals. Figure 4e takes the MPE variance out, in order to compare the refined methods. As we can see, cMPE has the highest variance, followed by rcDPE, rcMPE, and rcDPE, in that order. The black dotted line represents the theoretical variance for cMPE and cDPE. We can see here that cDPE actually performs sightly better than expected. Finally, Figure 4f shows only the best performing methods, rcMPE and rcDPE. From Figure 4f, we can see that rcDPE is not only the method with the lowest variance (three orders of magnitude less than the variance of MPE), but the only one which does not increase with scale. Moreover, rcDPE variance runs along the predicted theoretical value from Equation (Equation 35). The fact that rcMPE still presents a variance increase with respect to *m* is explained by the artifact cross-correlations, which we know from Section 3.1.3, are scale-dependent.

Therefore, from this initial experiment, we observe rcDPE is the only method which minimizes bias and variance, as well as having the desirable property of not introducing scale-dependent behaviors. This is particularly useful if we want to avoid any distortion due to the entropy calculation algorithms. From the statistical perspective, rcDPE is the method we recommend for PE analysis.

The rcDPE main constraint does not come from its statistical behavior, but from the frequency properties. In contrast to multiscaling, the downsampling procedure does not add any kind of filter to the signal, which renders the method particularly prone to aliasing effects. This problem is not evident from the study of uncorrelated noise, and will be discussed the next section.

### 4.2. Applications on Fault Bearing

Our goal in this section is to showcase the application of the entropy methods here discussed to classify the base and the fault signals. We are especially interested in the performance of rcMPE and rcDPE, since these two methods have the least amount of bias and variance. Therefore, we will apply MPE (for reference), rcMPE, and rcDPE algorithms in the bearing fault data set provided by [14], using the parameters d={3,4,5,6} and m=1,⋯,20 (higher values of *m* do not provide additional information in this case). For simplicity, we will set the downsampling parameter as τ=m.

The bearing fault data set [14] consist of signals from a bearing test rig, with a load of 270 lbs, input shaft rate of 25 Hz, and a sampling rate of $f_{s}=$ 97,656 Hz for 6 s (N= 585,936). The data set has three signals at baseline conditions (labeled “base”), and three signals with outer race fault conditions (labeled “fault”). A further explanation of the data set is available at [14]. Some example periodograms are shown in Figure 5.

Before going further, we should revisit the problem of aliasing. As pointed by [8], a pure downsampling method is particularly prone to aliasing frequencies. This is particularly notorious at high time scales, where the Nyquist frequency gets reduced by a factor of 1/τ, as seen in Figure 5. Therefore, we will also perform an anti-aliasing low-pass filtering before using rcDPE, with a cutoff frequency of $f_{N,m}=f_{s}/\left(2\tau \right)$. Even if the analysis of aliased signals is not useful in practice, we will keep the unfiltered (rcDPE) procedure, for the purpose of comparison. We do not perform this filter preprocessing on MPE and rcMPE, since the algorithms perform a moving average filter [18] as part of their procedure.

As our last steps, we will perform an exploratory analysis in order to select the values of *m* (or τ) in which the difference between classes base/fault are most evident. The results of this exploration are shown on Figure 6. Based on these scale selection, we will perform normality and homoscedasticity tests in order to select and perform the appropriate statistical factor analysis.

We observe in Figure 6a,b the average entropy values for base and fault signals, respectively, for dimension d=3. The base signals present lower entropy values at m=1, which suggest more regular behavior than the fault signals. As we increase time scale, base signals quickly approach maximum entropy, while the fault signals still preserve some long-range regularities. The notable exception to this is the unfiltered rcDPE method, which presents a similar high entropy for both base and fault signals, at almost all time scales used. Figure 6c,d show the standard deviation measured from the base and fault signals. Here, we do not observe scale-dependent effects on the entropy measured, since the sample size N≈ 585,000 is high compared to the dimensions used (N≫d!). We observe the base signals to have a significant lower variation than the fault set, which is to be expected from bearings in optimal conditions. We can also observe, in particular from Figure 6d, that the unfiltered rcDPE shows the least variation among all methods. The filtered rcDPE, on the other hand, presents a similar curve as the MPE and rcMPE methods, both in the mean entropy and its standard deviation. This strongly suggest that the structures we observe from MPE, rcMPE, and filtered rcDPE are sensitive to the frequency content. Moreover, the fact that the unfiltered rcDPE leads to relatively uniform entropy readings does not come from randomness, but from aliasing. Even though the unfiltered rcDPE has the lowest overall standard deviation, the entropy values themselves are unexpectedly high due to effects not related to the statistical properties of the problem. Therefore, proper filtering is necessary for multiscale entropy analysis.

Since our main goal in this section is to differentiate between base and fault signals, we should also take a closer look at the mean entropy difference ΔH, for d=3. Figure 7 shows this measurement with respect to time scale for each entropy method. First, we can observe that, at m=1, all methods can differentiate between base and fault signals. Except for unfiltered rcDPE, all methods increase their error for high time scales. Although the difference between mean entropy measurements is not high at m=4, this is the time scale with the minimum overall standard deviation for all cases. Therefore, for the purposes of classification, we will choose m=4. Furthermore, even if the unfiltered rcDPE offers significantly less error overall, the reduced ΔH, product of aliasing, makes this approach unsuitable for this classification problem. Therefore, we will include unfiltered rcDPE results just for reference.

Now, we proceed to make a factor analysis for the entropy results at m=4. We are interested in the type (base/fault signals), method (MPE, rcMPE, rcDPE, and filtered rcDPE), and dimension (d={3,4,5,6}). First, we performed the Shapiro–Wilk normality test and the Levene’s homoscedasticity test. We found no statistically significant evidence of deviation from normality or equal variance. Therefore, we performed a three-way ANOVA test for the aforementioned factors. We found all main factors and interactions to be statistically significant with α=0.05.

The presence of heavy interaction effects render the interpretation difficult, but we can still comment on the overall effects using Figure 8. Here we can first observe that dimension d=3 is the most suited for the classification of this particular data set, since all the entropy measurements are different, statistically speaking. As we increase in dimension, only the rcDPE methods remain significant. At d=6, only the filtered rcDPE method still present significantly differences in entropy between base and fault signals. Figure 8 also shows that filtered rcDPE has a notoriously higher difference between mean entropy than the other methods. This fact implies a better classification power using this technique.

Therefore, the best parameters for classification in this particular data set are m=4, d=3 (since higher dimensions do not bring better performance), using the rcDPE method with a low-pass frequency filter adjusted with scale. These results are by not means universal, since statistical and frequency characteristics of the phenomenon play a major role in the overall entropy behavior. Nonetheless, this example shows how the multiscale permutation entropy statistical development can be used for the purpose of classification, as well as highlight the advantages of the downsampling techniques combined with the appropriate filtering.

## 5. Conclusions

In the present work, we addressed the characterization of the composite MPE (cMPE) [8] and refined composite MPE (rcMPE) [9] statistical properties, with a special emphasis on an explicit, theoretical expected value, bias, and variance. This allowed us to better understand the improved results obtained [8,9], respect to the original MPE. We also identified the problem of artifact cross-correlation, inherent of the composite coarse-graining procedure, which leads to redundant information, and thus, an sub-estimation of the real entropy content within a signal. In order to address this problem, we revisited and re-framed the downsampling MPE procedure [20]. On top of this, we proposed a new PE method named Refined Composite Downsampling Permutation Entropy (rcDPE), which combines the composite downsampling approach with the refinements proper of the rcMPE technique. We also tested the performance of these techniques by measuring the expected entropy result and variance from uncorrelated noise. Finally, we showcased an example of practical application on real bearing fault recordings [14], for the purposes of classification.

Results from the analysis of uncorrelated noise showed the downsampling methods present less variance compared to the corresponding entropy methods found in the literature. In particular the rcDPE method is the best performing algorithm in terms of reduced bias and variance. Furthermore, rcDPE is the only method where we found no time scale dependency, in accordance to the theoretical predictions.

When applied to bearing fault recordings, we found the optimal scale and dimension to maximize the statistical difference in entropy between baseline operation and faulty components. The MPE, rcMPE and rcDPE were selected for comparison. Although it is possible to find significant results with the multiscaling procedures, we found the rcDPE, with an appropriate anti-alias low-pass filter, to outperform the MPE and rcMPE. The use of a naïve, non-filtered rcDPE, although minimizing the variance of the entropy results at all scales, also over-estimates the entropy values due to white noise spectrum occurring at high frequencies. This decreases the signal-to-noise ratio, increasing the signal’s randomness. The anti-aliasing low-pass filter with rcDPE presents a better option in terms of statistical efficiency and proper management of aliasing frequencies.

It is notable to find a procedure, such as rcDPE, which can maintain a constant minimum bias and variance across all time scales. This fact actually implies that the signal length constraint—the most cited limitation—is greatly diminished, specially at high time scales, where the downsampled signals get short and the estimations high on error. Since the results match the theoretical calculations we provide for the rcDPE estimator’s moments, we can guarantee stable results for short signals and high time scales.

With this work, we expect to provide a greater picture—through theoretical development —regarding the expected statistical properties of multiscale permutation entropy methods. The proposed downsampling permutation entropy techniques allow researchers to achieve higher precision in the entropy estimation, by avoiding artifacts introduced by the methods themselves. These techniques cannot be applied blindly, since the method is sensitive to frequency effects. Consequently, the appropriate application of signal pre-processing is needed to extract results closer to the real information content of the phenomenon in question.

## Figures and Tables

**Figure 1 entropy-23-00030-f001:**
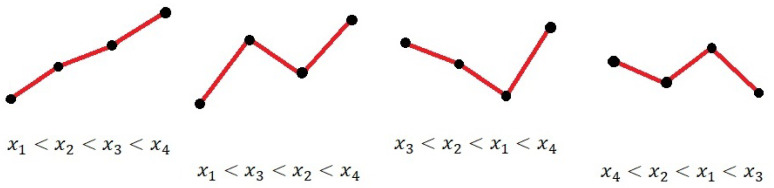
Ordinal pattern examples for τ=1. The figures represent discrete data points from a uniformly sampled signal. There are 24 possible patterns for d=4.

**Figure 2 entropy-23-00030-f002:**
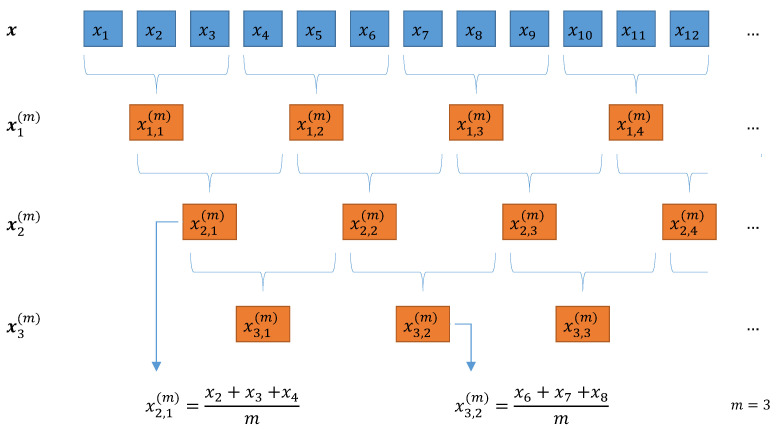
Schematic representation of the composite coarse-graining procedure at m=3: we segment the original signals in nonoverlapping segments of size *m*; if we shift the initial position, we can build *m* different coarse signals $x_{1}^{\left(m\right)}$, $x_{2}^{\left(m\right)}$, and $x_{3}^{\left(m\right)}$.

**Figure 3 entropy-23-00030-f003:**
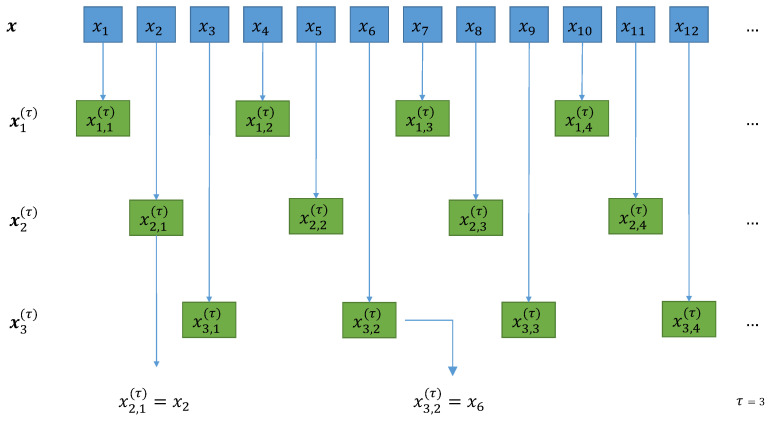
Schematic representation of the composite downsampling procedure at τ=3: we downsample the original signals by taking data points that are τ spaces apart; if we shift the initial position, we can build τ signals. The present downsampling signals share no mutual data points between them.

**Figure 4 entropy-23-00030-f004:**
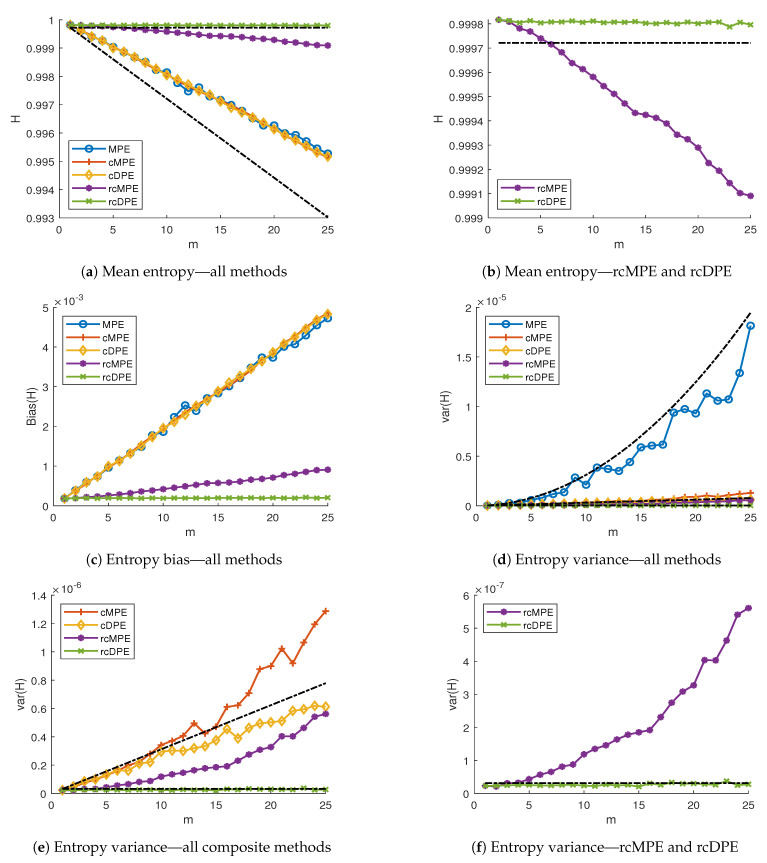
Permutation entropy methods applied to 500 white Gaussian noise signals at *μ* = 0 and *σ*^2^ = 1, with signal length *N* = 1000 and *m* = 1, …, 25. (**a**) shows the mean PE value of all methods, respect to
time scale *m* = *τ*. (**b**) showcases the mean rcMPE and rcDPE methods only. (**c**–**f**) show the comparisons
between the variance each method, where (**c**) present all methods, (**d**) present all the composite and refined
composite methods, and (**f**) shows rcMPE and rcDPE only. Black dotted lines show the theoretical values
predicted from Section 3.

**Figure 5 entropy-23-00030-f005:**
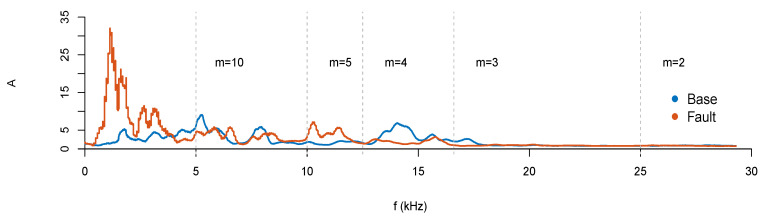
Periodogram of bearing fault tests. The blue curve shows a sample baseline signal, while the red curve shows a bearing with outer race fault conditions. With a sampling frequency of $f_{s}\approx 100\mathrm{kHz}$, the signal at scale m=1 (or τ=1 for the downsampling procedure) has a Nyquist frequency of $f_{N}\approx 50\mathrm{kHz}$. Each increasing time scale reduces $f_{N}$ by a factor of 1/m (or 1/τ), which can quickly introduce aliasing effects.

**Figure 6 entropy-23-00030-f006:**
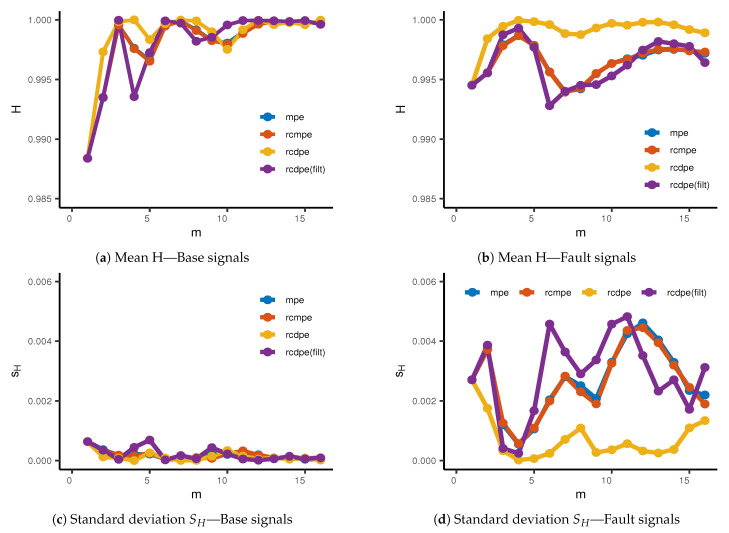
Mean entropy values for (**a**) Base and (**b**) Fault signals, for the MPE, rcMPE, rcDPE (unfiltered), and
rcDPE (filtered) methods. (**c**,**d**) present the standard deviation *S_H_* for Base and Fault signals, respectively.

**Figure 7 entropy-23-00030-f007:**
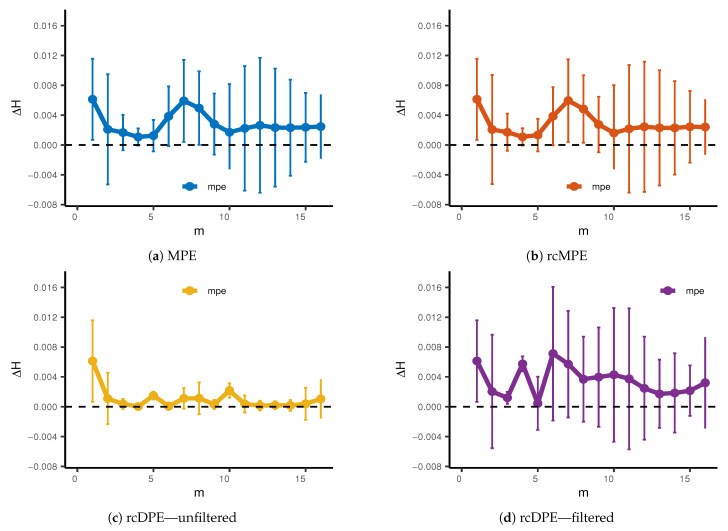
Difference between Base and Fault mean entropy Δ*H* for (**a**) MPE, (**b**) rcMPE, (**c**) rcDPE unfilteredand (**d**) rcDPE filtered. Error bars are computed using *S*_Δ*H*_ = ±1.96$\sqrt{S_{H,Base}^{2}+S_{H,Fault}^{2}}$, assuming a normal distribution with *α* = 0.05 significance level.

**Figure 8 entropy-23-00030-f008:**
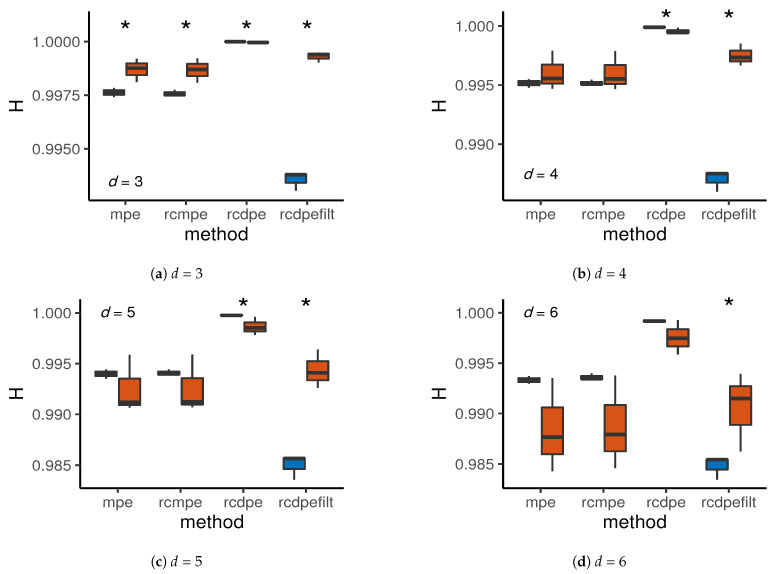
Base-fault classification for scale *m* = *τ* = 4 for MPE, rcMPE, and rcDPE (unfiltered and filtered), for dimensions *d* = {3,4,5,6}. Asterisks mark statistically significant differences between signal types, with *α* = 0.05. (**a**) *d* = 3; (**b**) *d* = 4; (**c**) *d* = 5; (**d**) *d* = 6.

## Data Availability

Not applicable.

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
