# Peer review of "Improvement of Statistical Performance of Ordinal Multiscale Entropy Techniques Using Refined Composite Downsampling Permutation Entropy"

_entropy, 2020, doi:10.3390/e23010030_

Round 1
Reviewer 1 Report
This paper presents nice improvements in the study of entropy methods to analyse time series. In particular the authors consider permutation entropy and its multiscale version.
I think that the paper is well written and the results are well presented. My opinion is that the paper is ready for publication on this journal.
Author Response
Thank you very much for your review and comments. We made some small modifications to improve the reading flow. You can find the corrected pdf attached to this message, with comments and corrections in blue.
Nonetheless, here is a list of the modifications, for reference:
- Abstract complete revision.
- Section 2.3, line 116
- Corrected "equation" instead of "equ." for consistency.
- Eq. 24.
- Correction. Should read \approx instead of <=
- Eq. 25.
- Correction. Should read \approx instead of >=
- Section 3.1 line 232
- Changed "original signal" instead of "signals in question", to improve reading flow.
- Section 3.2.3, line 311
- Changed N^+ into stylized N, to point the domain of natural positive numbers, and not the signal length N.
- Increased resolution in figures 5 & 6.
- Section 4.1 line 357
- Changed first instance of "properties" to "behavior", to improve grammar.
- Section 4.2, line 412
- Deleted word "only".
- Moved figures 5, 6, 7, & 8 closer to their respective references in the main text.
- Conclusions, line 450
- Added "time scale" instead of "scale", for clarity.
- Conclusions, line 455
- Added comma (in blue).
- Conclusions, line 456
- Added diaeresis to the word "naïve".
Once again, thank you very much.

Reviewer 2 Report
The manuscript is well-written with interesting contribution. The technical approach and the results are sound. Therefore, I recommend to accept this manuscript after correcting the minor issues.
1) The contribution should be highlighted in the abstract to improve the readability of the paper.
2) The figure quality should be improved, Fig 2 and 3 are not clear.
3) Some minor typesetting problem should be further considered as the font of in the figures are not consistent.
Author Response
Thank you very much for your review and suggestions. We addressed your comments as follows:
1) The contribution should be highlighted in the abstract to improve the readability of the paper.
The abstract has been rewritten, emphasizing the contributions of our work. Please let us know if the new text does indeed reflect the paper's content better.
2) The figure quality should be improved, Fig 2 and 3 are not clear.
The figures were modified in higher resolution
3) Some minor typesetting problem should be further considered as the font of in the figures are not consistent.
Similarly, we modified the font inside the figures, to match the same font as the main text equations.
Additionally, we made some small corrections through the main text. We include the corrected pdf document in this message, with the appropriate comments. All changes are highlighted in blue font.
Additionally, here is a complete list of all these changes:
- Abstract complete revision.
- Section 2.3, line 116
- Corrected "equation" instead of "equ." for consistency.
- Eq. 24.
- Correction. Should read \approx instead of <=
- Eq. 25.
- Correction. Should read \approx instead of >=
- Section 3.1 line 232
- Changed "original signal" instead of "signals in question", to improve reading flow.
- Section 3.2.3, line 311
- Changed N^+ into stylized N, to point the domain of natural positive numbers, and not the signal length N.
- Increased resolution in figures 5 & 6.
- Section 4.1 line 357
- Changed first instance of "properties" to "behavior", to improve grammar.
- Section 4.2, line 412
- Deleted word "only".
- Moved figures 5, 6, 7, & 8 closer to their respective references in the main text.
- Conclusions, line 450
- Added "time scale" instead of "scale", for clarity.
- Conclusions, line 455
- Added comma (in blue).
- Conclusions, line 456
- Added diaeresis to the word "naïve".
Once again, thank you very much for your input. We are open for any additional discussion regarding this work.
